# A pilot study using hospital surveillance and a birth cohort to investigate enteric pathogens and malnutrition in children, Dili, Timor-Leste

Danielle M. Cribb[1]*, Nevio Sarmento[2,3], Almerio Moniz[1], Nicholas S. S. Fancourt[2], Kathryn Glass[1], Anthony D. K. Draper[1,2,4], Joshua R. Francis[2], Milena M. Lay dos Santos[3], Endang Soares da Silva[3], Benjamin G. Polkinghorne[1], Virginia de Lourdes da Conceição[2,3], Feliciano da Conceição[5], Paulino da Silva[2], Joanita Jong[5], Martyn D. Kirk[1], Samantha Colquhoun[1]

1 National Centre for Epidemiology and Population Health, Australian National University, Canberra, Australian Capital Territory, Australia, 2 Global and Tropical Health Division, Menzies School of Health Research, Charles Darwin University, Darwin, Northern Territory, Australia, 3 Timor-Leste Ministry of Health, Dili, Timor-Leste, 4 Northern Territory Centre for Disease Control, Northern Territory Government Department of Health, Darwin, Northern Territory, Australia, 5 Ministry of Agriculture and Fisheries, Dili, Timor-Leste

* Danielle.Cribb@anu.edu.au

**Data Availability Statement:** All relevant data are within the paper and its Supporting Information files.

## Abstract

In low-to-middle-income countries (LMICs), enteric pathogens contribute to child malnutrition, affecting nutrient absorption, inducing inflammation, and causing diarrhoea. This is a substantial problem in LMICs due to high disease burden, poor sanitation and nutritional status, and the cyclical nature of pathogen infection and malnutrition. This relationship remains understudied in Timor-Leste. In our pilot study of enteric pathogens and malnutrition in Dili, Timor-Leste (July 2019–October 2020), we recruited 60 infants in a birth cohort from Hospital Nacional Guido Valadares (HNGV) with up to four home visits. We collected faecal samples and details of demographics, anthropometrics, diet and food practices, and animal husbandry. Additionally, we collected faecal samples, diagnostics, and anthropometrics from 160 children admitted to HNGV with a clinical diagnosis of severe diarrhoea or severe acute malnutrition (SAM). We tested faeces using the BioFire® FilmArray® Gastrointestinal Panel. We detected high prevalence of enteric pathogens in 68.8% (95%CI 60.4–76.2%) of infants at home, 88.6% of SAM cases (95%CI 81.7–93.3%) and 93.8% of severe diarrhoea cases (95%CI 67.7–99.7%). Diarrhoeagenic *Escherichia coli* and *Campylobacter* spp. were most frequently detected. Pathogen presence did not significantly differ in birth cohort diarrhoeal stool, but hospital data indicated associations between *Salmonella* and *Shigella* and diarrhoea. We observed wasting in 18.4% (95%CI 9.2–32.5%) to 30.8% (95%CI 17.5–47.7%) of infants across home visits, 57.9% (95%CI 34.0–78.9%) of severe diarrhoea cases, and 92.5% (95%CI 86.4–96.2%) of SAM cases. We associated bottle feeding with increased odds of pathogen detection when compared with exclusive breastfeeding at home (OR 8.3, 95%CI 1.1–62.7). We detected high prevalence of enteric pathogens and signs of malnutrition in children in Dili. Our pilot is proof of concept for a study to fully explore

**Funding:** This study was funded by the Australian Centre for International Agricultural Research (project number LS/2018/184v1). DMC is funded by an Australian Government Research Training Program (AGRTP) Scholarship. MDK is supported by a National Health and Medical Research Council fellowship (GNT1145997). The funder (ACIAR) did not have a role in study design, data collection and analysis, decision to publish, or preparation of the manuscript.

**Competing interests:** The authors have declared that no competing interests exist.

the risk factors and associations between enteric pathogens and malnutrition in Timor-Leste.

## Introduction

Childhood undernutrition is a major challenge for many countries and is associated with high levels of morbidity and mortality [1]. In 2020, 22.0% of children aged under five years worldwide suffered from stunting (low height for age) and 6.7% suffered from wasting (low weight for height) [2]. Children in low- and middle-income countries (LMICs) are disproportionately impacted by stunting (38.8%) and wasting (12.9%) [3]. Undernutrition can lead to cognitive, physical, and metabolic developmental issues, potentially reducing physical and intellectual ability and economic productivity in adulthood [1].

Socioeconomic and environmental factors contribute to child undernutrition, including poor maternal health and education, early termination of breastfeeding, low household wealth, and unimproved sanitation [3, 4]. There is growing evidence for an association between infection with enteric bacteria, viruses, and parasites and child stunting and wasting, particularly in LMICs [5, 6]. Enteric bacterial pathogens contracted early in life may damage the lining of the small and large intestines, reducing the absorption of nutrients from food and promoting chronic inflammation, leading to undernutrition and a compromised immune system [5, 6]. Such enteric pathogens potentially involved in malnutrition include enterotoxigenic *Escherichia coli* (ETEC), *Shigella*, and *Campylobacter* species [7–9].

*Campylobacter* is the most common cause of foodborne bacterial gastroenteritis globally, causing an estimated 96 million cases and >21,000 deaths annually [10]. In high-income settings, infections are often sporadic in nature and largely attributed to consuming undercooked chicken [11]. In low-income settings, *Campylobacter* is frequently endemic and while still largely attributed to poultry, infection determinants are from more varied environmental and zoonotic sources [11, 12]. Recent studies have further implicated *Campylobacter* in child malnutrition. A Peruvian cohort exhibited poor weight gain with infection and marginal poor linear growth with symptomatic infection [7, 8]. Similarly, 84.9% of the multisite Malnutrition and Enteric Disease Study (MAL-ED) cohort presented with a *Campylobacter*-positive faecal sample by one year of age, with those diagnosed with multiple *Campylobacter* infections being more likely to exhibit poor linear growth [12].

Timor-Leste is one of the lowest-income countries in Asia and has a population of 1.34 million circa 2022 [13, 14]. The 2016 Demographic and Health Survey (DHS) estimated that the proportion of children with stunting was 46.5%, wasting 24.2% and underweight (low weight for age) 40.4% [15]. Diarrhoeal disease is prevalent in Timor-Leste with 10.7% of children represented in the DHS presenting with diarrhoea within the two weeks preceding the survey, peaking at 17.8% in children aged 12–23 months [16].

While malnutrition is a known issue in Timor-Leste, the prevalence of enteric pathogen infections in children is not well understood. We conducted a two-arm pilot investigation to establish whether a high prevalence of enteric pathogens and malnutrition coexists in children in Timor-Leste from an early age, and to examine the practicality of a longitudinal birth cohort and hospital surveillance study. With the birth cohort, we aimed to i) estimate the proportion of moderately to severely malnourished children (with z-scores of ≤-2 for stunting, wasting, or underweight) during the first 12 months of life; ii) determine the prevalence of enteric pathogens in stool samples using polymerase chain reaction (PCR), identify those with multiple pathogen detections, identify the most common pathogens detected, and identify any

differences in pathogen detections between diarrhoeal and non-diarrhoeal stool; and iii) identify household, environmental, and One Health risk factors associated with enteric pathogen infections to identify potential intervention targets. For the hospital surveillance cohort, we aimed to iv) quantify malnourishment using different metrics in children diagnosed with severe acute malnutrition (SAM) or severe diarrhoea; and v) estimate the prevalence of enteric pathogens in stool samples, identify those with multiple pathogen detections, identify the most common pathogens, and identify any differences in pathogen detections between diarrhoeal and non-diarrhoeal stool in children admitted with these conditions. This investigation formed the basis of a larger study to understand the impact of enteric pathogens on malnutrition.

## Methods

This study was carried out at the national referral hospital in Dili, Timor-Leste, Hospital Nacional Guido Valadares (HNGV). HNGV has a catchment population of 500,000 people with approximately 4,500 births per year.

### Longitudinal birth cohort

**Study setting and recruitment.** We recruited newborns into the longitudinal birth cohort between July and September 2019. Newborn infants were eligible for inclusion if they were born at ≥38 weeks gestation at HNGV. Children were excluded if they were reported to live outside a 10-kilometre radius of HNGV or if their mothers were not Timorese citizens. The sample size of 60 infants was established with an estimated dropout rate of 33% (leaving an effective sample size of 40 infants) [17]. The local research team approached parents in the maternity unit at HNGV after delivery and prior to discharge. Parents provided informed consent to enrol their infants. Home visits were initially scheduled at two-month intervals from birth until six months of age. However, due to local flooding and COVID-19 lockdowns, visits occurred when infants were approximately one, four, seven, and 12 months of age (S1 Table). As this was a proof-of-concept pilot study, we did not conduct any additional recruitment to account for loss to follow up.

**Equipment and questionnaire.** The field study team were trained in Good Clinical Practice for research by the project lead [18]. The local study team were trained to use a calibrated portable baby scale and a standardised mat for measuring weight (Seca 334 Baby Scales) and height (Seca 210 Mobile Measuring Mat for Babies and Toddlers). All infants were weighed and measured using a standardised procedure [19]. In addition, the team interviewed parents using a standardised questionnaire at each visit to investigate feeding practices, household size, water, sanitation, and hygiene (WASH) practices and accessibility, and household animal ownership and containment (S1 File) [16]. The questionnaire was based on the 2016 DHS approved for use by the Timor-Leste Ministry of Health. All data were collected on an android tablet using Research Electronic Data Capture (REDCap) [20, 21], synchronised online, and stored on a secure server hosted at the Australian National University (ANU).

**Specimen collection.** Faecal samples were collected in sample pots from all infants at the time of the field visit or in the following days and transported to the lab in an insulated United Nations Children's Fund container. Parents and/or guardians were provided with instructions in their native language, Tetum, to ensure samples were collected on the day of or the evening before the home visit and stored in a cool location prior to transport. Our team provided appropriate equipment including gloves, spatula, specimen container, and clinical waste bag. Samples were transported to the National Health Laboratory (NHL) for stool assessment and PCR testing.

## Hospital surveillance

**Study setting and recruitment.**   We recruited children admitted to HNGV with diarrhoea with severe dehydration (DWSD) or dysentery (collectively referred to as severe diarrhoea), or SAM between October 2019 and July 2020 to investigate enteric pathogens in children with these conditions in Dili. Children were eligible for inclusion if they resided within Dili and if they met the adapted World Health Organization (WHO) criteria for diagnosis of SAM, DWSD and/or severe dysentery [22]. Children were excluded if they were aged over five years at time of hospital admission. Our team attempted to recruit every eligible child that presented to the hospital until we reached the target sample size.

**Specimen and information collection.**   Participants were identified for eligibility by pae-diatricians during the daily paediatric ward round. Ward nurses collected faecal samples from eligible children during routine clinical treatment. No specific consent was sought regarding the testing of stool as this was part of routine public health follow-up and no identifying infor-mation was collected. No contact was initiated by researchers with these children or their fami-lies and only specimen date, age, sex, height, weight and middle-upper arm circumference were recorded. The study team transported the samples to the NHL for stool assessment and PCR testing.

## Pathogen detection

Laboratory staff processed samples from the birth cohort and hospital surveillance cohort using the same method. The NHL received faecal samples and performed visual inspections of the stool, routine microscopy including motility, and Gram stain morphology. In line with the standardised protocol, samples were stored either at room temperature (15–25˚C) or refriger-ated (2–8˚C) for up to four days [23]. Laboratory staff transferred approximately 1g of stool to a screw-cap tube filled with 15ml of Cary Blair liquid medium (Remel, Lenexa, KS, USA) which was shaken until homogenous [24]. The NHL used the BioFire® Gastrointestinal Panel (GI) Kit for in-house PCR testing on all faecal samples from birth cohort and hospitalised chil-dren against 22 gastrointestinal pathogens, namely *Campylobacter* (*C. jejuni*/*C. coli*/*C. upsa-liensis*), *Clostridioides* (*Clostridium*) *difficile* (toxin A/B), *Plesiomonas shigelloides*, *Salmonella*, *Yersinia enterocolitica*, *Vibrio* (*V. parahaemolyticus*/*V. vulnificus*/*V. cholerae*), Enteroaggrega-tive *E. coli* (EAEC), Enteropathogenic *E. coli* (EPEC), ETEC lt/st, Shiga-like toxin-producing *E. coli* (STEC) stx1/stx2, *E. coli* O157, *Shigella*/Enteroinvasive *E. coli* (EIEC), Adenovirus F40/41, Astrovirus, Norovirus GI/GII, Rotavirus A, Sapovirus (I, II, IV, and V), *Cryptosporidium*, *Cyclospora cayetanensis*, *Entamoeba histolytica*, and *Giardia lamblia*. Samples were prepared using a 30-test kit (RFIT-ASY-0116, bioMérieux Australia PTY LTD, AU) according to manu-facturer's instructions [23]. We considered samples with a positive PCR result for two or more of the screened pathogens to have multiple pathogens detected. Samples positive by BioFire® PCR for selected enteric bacteria (*Campylobacter*, *Salmonella* and *Shigella*/EIEC) were reflex cultured on appropriate selective media and an animal health team returned to the household to collect animal samples.

## Data analysis

All data were cleaned and analysed using R statistical software [25]. We adopted different anal-ysis methods for each cohort based on whether they included repeated observations. Any par-ticipants with missing data were only excluded from analyses that required the missing data.

**Longitudinal birth cohort.**   We converted anthropometric measures to weight-for-height (WHZ), height-for-age (HAZ), and weight-for-age (WAZ) Z scores using R package "zscorer" [26] and mapped these against the WHO child growth standards [27, 28]. While all three

metrics were calculated, we focused on WHZ as an outcome variable as wasting can be a precursor to other signs of malnutrition, including stunting [29]. We created categorical variables for age (three-month intervals), pathogen presence (one, two, or three or more pathogens present), and categorical risk factor variables like household size and number of animals (grouped into intervals of five). We created binary variables for the outcomes of moderate to severe wasting ($\leq$-2 WHZ or >-2 WHZ) and pathogen detection (any pathogens detected or no pathogens detected) [30]. We performed Fisher's exact and Chi-square tests to analyse differences between pathogen group (bacteria, viruses, or parasites), age group, and home visit. We calculated odds ratios (OR), controlling for age, sex, and season, using generalised estimating equations (GEEs) to account for repeated observations with the R package "geepack", where appropriate [31]. GEEs enable effective control for repeated measures by incorporating within-subject or within-cluster correlations, thereby providing robust analysis of longitudinal or clustered data [32]. GEEs were used to analyse risk factors against the outcomes of i) pathogen detection and ii) moderate to severe wasting. Additionally, we used GEEs to analyse differences in pathogen detections between diarrhoeal and non-diarrhoeal stool samples. GEE models used an exchangeable correlation structure and binomial distribution with logistic link. We used the "arsenal" package to calculate generalised linear models, where appropriate [33]. A $p$ value of $p < 0.05$ was considered significant for Fisher's and Chi-square tests, and an OR with confidence intervals (CI) that did not cross over one was considered significant for GEEs.

**Hospital surveillance.** We converted anthropometric measures to WHZ, HAZ, WAZ, and MUAC-for-age (MAZ) Z scores using "zscorer" [26]. We created categorical variables for age (yearly from birth until age five) and pathogen presence (one, two, or three or more pathogens present). We performed Fisher's exact, Chi-square, and Student's t-Test, where appropriate, to analyse differences between pathogens detected by age group and hospital diagnosis. We used a generalised linear model, adjusted for age group, sex, and season to analyse differences in pathogen detections between diarrhoeal and non-diarrhoeal stool samples. A $p$ value of $p < 0.05$ was considered significant.

## Inclusivity in global research

Additional information regarding the ethical, cultural, and scientific considerations specific to inclusivity in global research is included in the Supporting Information (S1 Checklist).

## Results

### Longitudinal birth cohort

We approached 68 families for the birth cohort and recruited 60 infants (32 males and 28 females). Reasons for exclusion included living outside of the study area ($n$ = 5) and no parental consent given ($n$ = 3). We retained 53.3% (32/60) of participants until the final home visit resulting in 161 completed visits. Reasons for non-completion throughout the study included loss to follow up, moving out of the study area, withdrawal by a parent, and child death (Fig 1). While the median infant WHZ at birth was 0.9 (interquartile range [IQR] -0.5 to 1.6), scores became negative across subsequent visits (S1 Table). WAZ showed similar decreases across visits while HAZ remained stable, indicating that poor weight gain may be a driving factor for malnutrition. Compared to the WHO Growth Standards, the birth cohort tracked the 50th percentile for stunting and between the 15th and 25th percentiles for underweight and wasting (Fig 2). During home visits, between 18.4% and 30.8% of infants recorded a WHZ consistent with diagnoses of moderate to severe wasting, while 0.0% to 9.4% of reported HAZ and 2.4% to 12.8% of reported WAZ were consistent with moderate to severe stunting and underweight, respectively (S1 Table) [22].

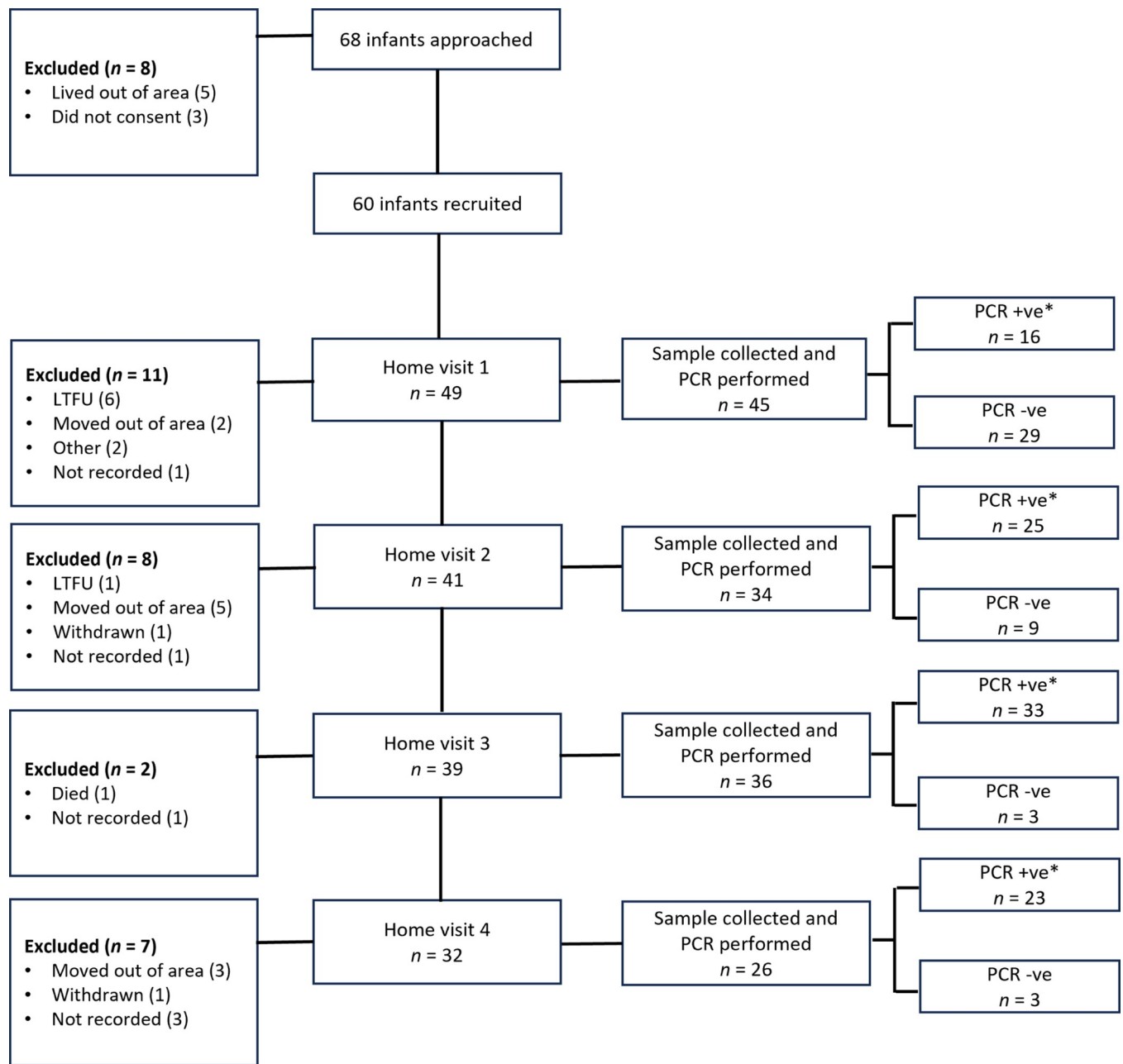

**Fig 1. Birth cohort recruitment and polymerase chain reaction (PCR) results in 60 infants, Dili, Timor-Leste, 2019–2020.** *See Table 1 for details on pathogens detected; +ve = positive, -ve = negative, LFTU = loss to follow up.

We collected 141 infant faecal specimens for PCR testing (87.6% sample collection rate). We detected at least one enteric pathogen in 68.8% (97/141; 95%CI 60.4%–76.2%) of samples, and of these 63.9% (62/97; 95% CI 53.5%–73.2%) were positive for multiple pathogens. We observed significant differences in pathogen detection across home visits. As visits progressed, both the number of pathogens detected within the cohort and the number of multiple pathogen detections in individual infants generally increased ($p < 0.05$; Table 1 and S1 Fig). While we did not detect a significant relationship between WHZ and pathogens detected for visits one, two, and four (linear model $p$ value $> 0.05$), we did detect a significant negative trend in

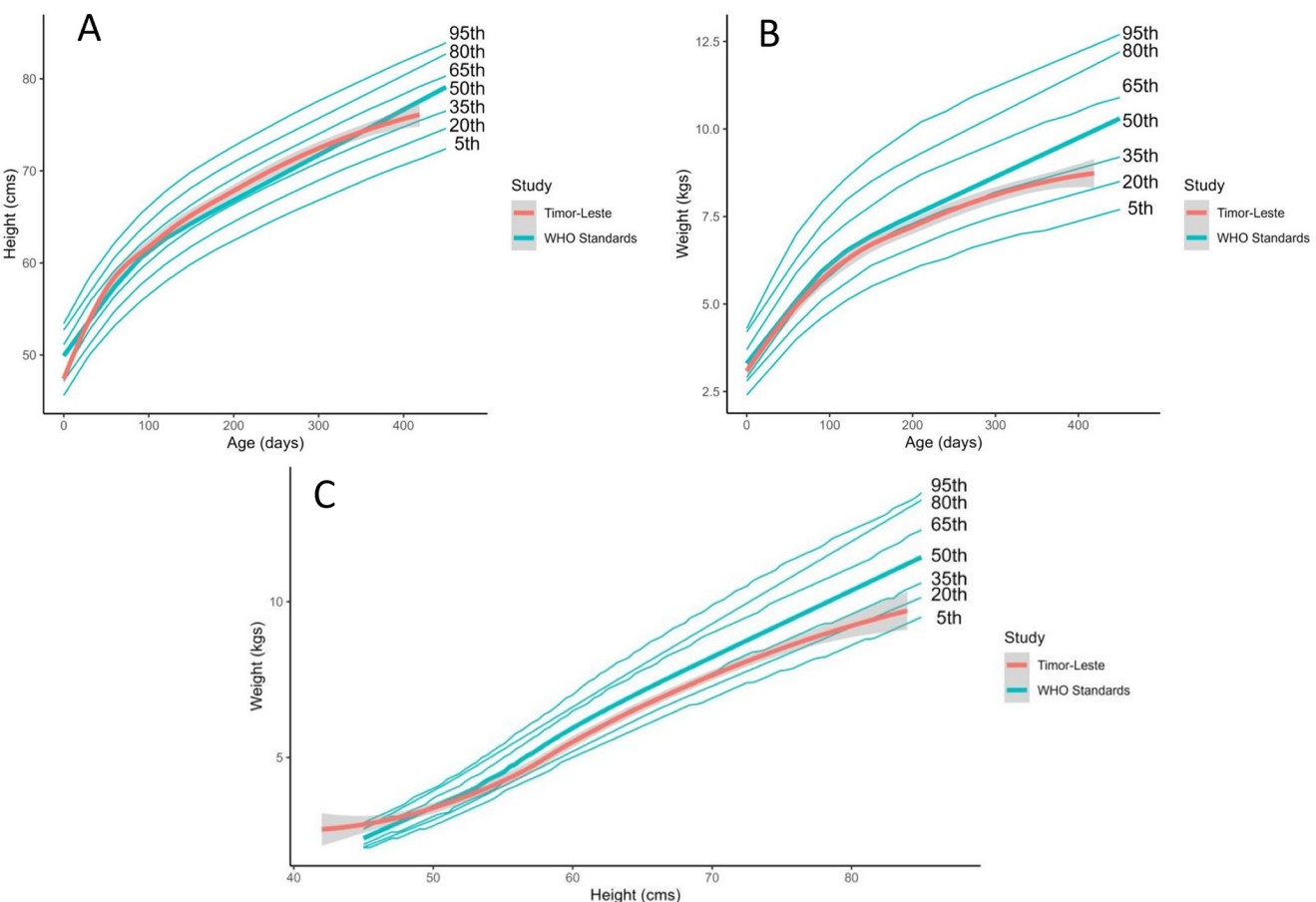

**Fig 2.** (a) Height-for-age World Health Organization (WHO) standards compared with Timor-Leste birth cohort, 2019–2020; (b) Weight-for-age WHO standards compared with Timor-Leste birth cohort, 2019–2020; (c) Weight-for-height WHO standards compared with Timor-Leste birth cohort, 2019–2020. WHO percentiles are annotated on figure. 50th percentile for WHO standards is presented in bold. Grey shading represents the 95% confidence interval for the Timor-Leste birth cohort.

visit three ($p$ value <0.01). However, we note that the small sample size and wide confidence intervals limit the interpretation of these results (Fig 3).

Diarrhoeagenic *E. coli* (DEC) strains (STEC, EPEC, EAEC, and ETEC) were the most common pathogens detected. Possible chronic carriage of EAEC was detected in six infants, with 96.0% of their samples PCR positive for the pathogen (six samples across four visits, 2S^#4). *Campylobacter* spp. were also frequently detected ($n$ = 15). We did not detect any significant differences in pathogens detected based on visually assessed diarrhoeal versus non-diarrhoeal stool samples (S2 Table). No human samples produced a positive result on culture, reflective of the low detection rate of *Salmonella* and *Shigella* and difficulties faced for culturing *Campylobacter*. The animal health team collected and cultured three animal samples from PCR-positive *Campylobacter* human samples from households with chickens and/or pigs. All three samples were culture-positive for *Campylobacter*.

Odds of pathogen detection increased with age, peaking in those aged over 12 months (OR 12.1, 95% CI 3.1–47.0) compared with children aged <3 months (S3 Table). We controlled for

**Table 1. Number and proportion of pathogens detected by multiplex PCR at each birth cohort home visit and stool presentation of samples, Dili, Timor-Leste, 2019–2020.**

| Pathogen | Visit 1 n (%) | Visit 2 n (%) | Visit 3 n (%) | Visit 4 n (%) | Total n (%) |
|---|---|---|---|---|---|
| **Bacteria** | | | | | |
| *Campylobacter* spp.*§ | 3 (6.7) | 1 (2.9) | 3 (8.3) | 8 (30.8) | **15 (10.6)** |
| *Clostridioides difficile* | 1 (2.2) | 4 (11.8) | 4 (11.1) | 4 (15.4) | **13 (9.2)** |
| *Salmonella*§ | 0 (0) | 0 (0) | 4 (11.1) | 0 (0) | **4 (2.8)** |
| Shiga toxin-producing *Escherichia coli* (STEC)§ | 0 (0) | 0 (0) | 0 (0) | 2 (7.7) | **2 (1.4)** |
| Enteroaggregative *E. coli*§ | 11 (24.4) | 21 (61.8) | 30 (83.3) | 15 (57.7) | **77 (54.6)** |
| Enteropathogenic *E. coli*§ | 3 (6.7) | 10 (29.4) | 21 (58.3) | 15 (57.7) | **49 (34.8)** |
| Enterotoxigenic *E. coli* lt/st§ | 1 (2.2) | 4 (11.8) | 12 (33.3) | 5 (19.2) | **22 (15.6)** |
| *Shigella*/Enteroinvasive *E. coli* | 1 (2.2) | 0 (0) | 1 (2.8) | 0 (0) | **2 (1.4)** |
| *Plesiomonas shigelloides* | 0 (0) | 0 (0) | 1 (2.8) | 0 (0) | **1 (0.7)** |
| *Vibrio* spp.† | 1 (2.2) | 0 (0) | 4 (11.1) | 0 (0) | **5 (3.5)** |
| **Viruses** | | | | | |
| Adenovirus F40/41§ | 0 (0) | 0 (0) | 2 (5.6) | 4 (15.4) | **6 (4.3)** |
| Norovirus GI/GII§ | 0 (0) | 3 (8.8) | 7 (19.4) | 2 (7.7) | **12 (8.5)** |
| Rotavirus A | 0 (0) | 3 (8.8) | 2 (5.6) | 4 (15.4) | **9 (6.4)** |
| Sapovirus§ | 0 (0) | 0 (0) | 4 (11.1) | 0 (0) | **4 (2.8)** |
| **Parasites** | | | | | |
| *Cryptosporidium* spp.§ | 0 (0) | 0 (0) | 0 (0) | 3 (11.5) | **3 (2.1)** |
| *Giardia lamblia* | 0 (0) | 0 (0) | 1 (2.8) | 0 (0) | **1 (0.7)** |
| **Total samples** | **45** | **34** | **36** | **26** | **141** |

*Campylobacter* spp. includes *C. jejuni, C. coli, and C. upsaliensis*. †*Vibrio* spp. includes *V. parahaemolyticus, V. vulnificus*, and *V. cholerae*. § indicates *p*-value of <0.05 for a significant change in pathogen detection across visits. N.B. percentages and counts do not equal total samples due to multiple pathogen detections in some samples.

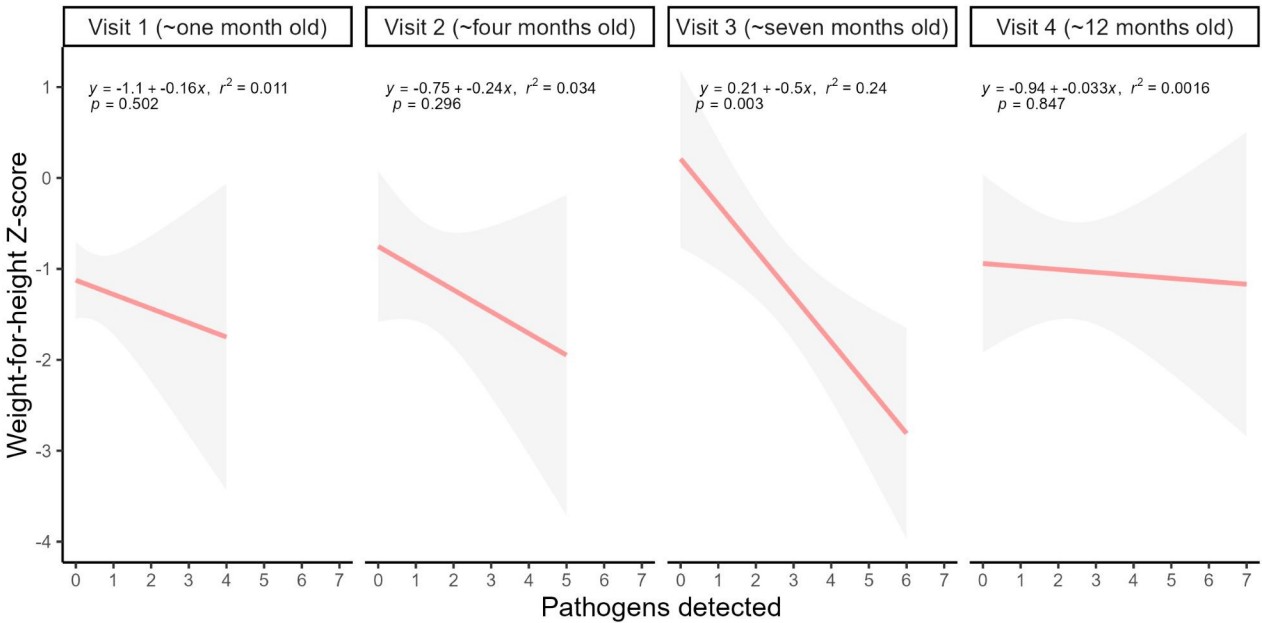

**Fig 3. Linear regression model for weight-for-height Z-score by number of pathogens detected for each birth cohort visit, Dili, Timor-Leste, 2019–2020.** 95% confidence intervals are indicated in grey shading for each line.

age, season, and repeated observations of individual participants in analysis of all additional risk factors. We found that exclusive bottle feeding was associated with increased odds of enteric pathogen detection (aOR 8.3, 95% CI 1.1–62.7) compared with breastfeeding. No other factors, including environmental and animal-related variables, were statistically significant. For the outcome of wasting, controlled for age, season, and repeated observations, we did not find any significant associations between risk factors (S4 Table). We note that the study is not sufficiently powered to demonstrate true non-associations.

## Hospital surveillance

We recruited 160 children in our hospital surveillance study arm. One child was excluded as they were aged above five years. Females represented 52.9% (83/157, missing data $n = 2$) of participants. We received PCR results from 148 participants as two patients were discharged before specimen collection and nine were missing laboratory PCR data.

The mean age at admission was 17.7 months (range 1.0–55.7 months, standard deviation (*SD*) ±11.4 months). Hospital doctors diagnosed 140 (88.1%, 95% CI 81.7–92.5%) children with SAM (132 with stool samples submitted to the NHL) and 19 (11.9%, 95% CI 7.5–18.3%) with severe diarrhoeal disease (13 with DWSD and six with dysentery, 16 with stool samples submitted to the NHL). The median WHZ for those with SAM (-3.9, IQR -4.7 to -3.0) was lower than those with severe diarrhoea (-2.6, IQR -4.2 to -1.1), however both groups exhibited moderate to severe wasting regardless of clinical diagnosis (S5 Table).

Of the 148 participants with complete multiplex PCR results, 88.6% (117/132; 95% CI 81.7–93.3%) of SAM participants were positive for at least one enteric pathogen, with 73.5% (97/132; 95% CI 65.0–80.6%) positive for multiple pathogens in a single sample. Of those with severe diarrhoea, 87.5% were positive for multiple pathogens in a single sample (14/16; 95% CI 60.4–97.8%). The type of pathogen detected varied by the age at admission with viral pathogens generally detected in children with a mean age of 13.8 months (*SD* ±8.1 months), while bacteria and parasites were detected in children with a mean age of 17.8 months (*SD* ±11.2 months), and 19.3 months (*SD* ±11.7 months), respectively. Age distribution remained similar regardless of diagnosis at admission (S6 Table).

The mean number of pathogens detected per individual was 2.8 (range 0–7). The most detected pathogens were DEC species followed by *Campylobacter* spp., *Shigella*/EIEC, and *Giardia lamblia* (Table 2). Four samples were successfully cultured for *Campylobacter* ($n = 2$) or *Salmonella* ($n = 2$). We detected more PCR-positive *Salmonella* (aOR 17.2, 95% CI 1.6–201.3) and *Shigella* (aOR 3.8, 95% CI 1.0–14.4) in visually assessed diarrhoeal stool samples compared with non-diarrhoeal samples but we note that the small sample size and wide confidence intervals limit the interpretation of these results (S7 Table).

## Discussion

Infants and young children from a birth cohort and hospitalisations due to SAM or severe diarrhoea exhibited high enteric pathogen presence, including the presence of multiple pathogens in single samples and signs of undernutrition from the first months of life. Ninety percent of children admitted to hospital for treatment of DWSD, dysentery, or SAM were positive for at least one enteric pathogen and 70% of infants visited at home also returned a positive PCR test for at least one enteric pathogen over the 12-month study. Pathogen detection generally increased with age within the birth cohort, likely influenced by exposure to new nutrition sources and environmental factors as noted in other LMIC cross-sectional studies [34, 35]. While the mechanisms of child malnutrition remain complex, our pilot study has provided important information on challenges faced and required modifications when conducting

**Table 2. Hospital surveillance data of diagnosis-specific pathogen detections and total pathogen detections from PCR testing, Dili, Timor-Leste, 2019–2020.**

| Pathogen | Pathogen detections in SAM cases n (%) | Pathogen detections in severe diarrhoea[‡] cases n (%) | All pathogen detections n (%) |
|---|---|---|---|
| **Bacteria** | | | |
| *Campylobacter* spp.* | 33 (25.0) | 3 (18.8) | 36 (24.3) |
| *Clostridioides difficile* | 6 (4.5) | 0 (0.0) | 6 (4.1) |
| *Plesiomonas shigelloides* | 1 (0.8) | 0 (0.0) | 1 (0.7) |
| *Salmonella* | 4 (3.0) | 1 (6.3) | 5 (3.4) |
| Shiga toxin-producing *Escherichia coli* (STEC) | 4 (3.0) | 0 (0.0) | 4 (2.7) |
| Enteroaggregative *E. coli* | 84 (63.6) | 10 (62.5) | 94 (63.5) |
| Enteropathogenic *E. coli* | 70 (53.0) | 10 (62.5) | 80 (54.1) |
| Enterotoxigenic *E. coli* lt/st | 41 (31.1) | 4 (25.0) | 45 (30.4) |
| *E. coli* O157 | 2 (1.5) | 0 (0.0) | 2 (1.4) |
| *Shigella*/Enteroinvasive *E. coli* | 23 (17.4) | 7 (43.8) | 30 (20.3) |
| *Vibrio* spp.[†] | 5 (3.8) | 3 (18.8) | 8 (5.4) |
| **Viruses** | | | |
| Adenovirus F40/41 | 3 (2.3) | 1 (6.3) | 4 (2.7) |
| Astrovirus | 6 (4.5) | 1 (6.3) | 7 (4.7) |
| Norovirus GI/GII | 12 (9.1) | 1 (6.3) | 13 (8.8) |
| Rotavirus A | 6 (4.5) | 1 (6.3) | 7 (4.7) |
| Sapovirus | 13 (9.8) | 1 (6.3) | 14 (9.5) |
| **Parasites** | | | |
| *Cryptosporidium* spp. | 17 (12.9) | 1 (6.3) | 18 (12.2) |
| *Giardia lamblia* | 31 (23.5) | 2 (12.5) | 33 (22.3) |
| *Cyclospora cayetanensis* | 12 (9.1) | 0 (0.0) | 12 (8.1) |
| *Entamoeba histolytica* | 0 (0.0) | 1 (6.3) | 1 (0.7) |
| **Total samples** | **132** | **16** | **148** |

*Campylobacter spp. includes C. jejuni, C. coli, and C. upsaliensis. †Vibrio spp. includes V. parahaemolyticus, V. vulnificus, and V. cholerae. ‡Severe diarrhoea includes diarrhoea with severe dehydration and dysentery diagnoses. PCR = polymerase chain reaction test. SAM = severe acute malnutrition. N.B. percentages and counts do not equal total samples due to multiple pathogen detections in some samples.

longitudinal research in Timor-Leste along with background data for a larger study exploring the relationship between enteric pathogens and malnutrition.

We detected a variety of enteric pathogens in children regardless of the presence of wasting and diarrhoeal disease. Across all faecal samples, *Campylobacter* spp., DEC strains EAEC, EPEC, and ETEC, *Shigella*/EIEC, norovirus, *Giardia*, and *Cryptosporidium* were most prevalent. This is consistent with results from the MAL-ED study [36] and the Global Enteric Multicenter Study (GEMS) [37]. Diarrhoeal faecal samples in our study frequently had DEC strains present, suggesting these may be driving disease presentation in combination with non-pathogenic factors. Pathogenic *E. coli* is commonly identified as a cause of both acute and persistent paediatric diarrhoea in low- and middle-income countries [38–41], particularly in malnourished children [42, 43]. Further, GEMS found that diarrhoea from typical EPEC and heat stable ETEC infections were associated with malnutrition and an increased risk of death in infants [41]. We detected possible chronic carriage or reinfection with EAEC in a small number of children in the birth cohort (*n* = 6). While persistent EAEC infections are known to be implicated in long-term diarrhoeal disease in children, this was not evident in our study [44–46]. We detected low prevalence of rotavirus in infants and children in both study arms. Routine vaccination for rotavirus commenced in Timor-Leste in late 2019, coinciding with our study [47].

While coverage rates are not reported for 2019, estimates from 2021 indicate approximately 80% coverage [48]. A study in Thai children receiving rotavirus vaccines found that the burden of diarrhoeal-related disease decreased after the rotavirus vaccine was introduced [49]. Children diagnosed with DWSD or dysentery in hospital-based surveillance had a high prevalence of DEC, *Campylobacter* spp. and/or *Shigella*, pathogens previously implicated with these conditions [50]. Despite high prevalence of certain pathogens, we did not detect any significant relationships between individual pathogens and wasting or diarrhoeal disease presentation. This is most likely due to the small sample size of our study but could also be related to multiple pathogen interactions or additional nutritional or environmental factors. Pathogen virulence may also play an important part in the distribution of pathogens between malnourished and/or diarrhoeal children versus those that are relatively healthy [51].

Most pathogens were recovered from faecal samples with limited or no evidence of diarrhoeal disease upon visual assessment in both study arms. While diarrhoeal disease is implicated in child malnutrition, subclinical infections with enteric pathogens may be more strongly linked to poor growth outcomes [12]. The MAL-ED study detected high prevalence of *Campylobacter* from an early age and a negative association between the presence of multiple pathogens and growth, particularly in asymptomatic cases [12]. Carriage of enteric pathogens like EAEC and *Campylobacter* can lead to enteric and systemic inflammation and malabsorption of nutrients resulting in malnutrition and faltered growth, particularly in prolonged cases or repeat infections [52–54]. The presence of enteric pathogens in asymptomatic children should be investigated further to work towards understanding the complex relationship between enteric pathogens and malnutrition [36].

Within Timor-Leste and neighbouring Southeast Asian countries, there is limited evidence of the prevalence of enteric pathogens both in clinical and environmental settings. A multisite study of children aged under five years conducted in Indonesia between 2009 and 2012 detected a high prevalence of rotavirus (37.4%), adenovirus (34.9%) and ETEC (10.0%) [55]. Another Indonesian case-control study of diarrhoeal disease conducted between 1997 and 1999 detected very low levels of enteric pathogens in paediatric patients using culture-based methods (i.e., less than 8.0% prevalence for total pathogens) [56], however, it is worth noting that culture-based methods can be lower in sensitivity than PCR-based methods.

We investigated risk factors in the home and surrounding environment in the birth cohort. We found that exclusive bottle feeding had an increased odds of pathogen detection compared to breast feeding. This is in line with a previous study in the Aileu District of Timor-Leste where early cessation of exclusive breastfeeding led to increased risk of exposure to contaminated water and food and unsatisfactory hygiene practices with subsequent diarrhoeal illness and malnutrition [4]. Further, although we did not find any significant animal-related risk factors, we were able to culture *Campylobacter* from animal stool in households where a human sample tested positive. Contact with animals and animal stool in the home environment could be an important transmission pathway for *Campylobacter* infections in children, with dogs and hens known to track human faecal material from open defecation into the home [4]. Flooding is common in Southeast Asia, particularly in Timor-Leste. Flooding events combined with ubiquitous pit latrines and improper garbage disposal provide ample opportunities for transmission of enteric pathogens in the environment [57]. A Nigerian study that tested groundwater sources for enteric pathogens found high levels of bacteria including *E. coli*, *Salmonella*, *Shigella*, and *Campylobacter* spp. in boreholes that likely weren't adequately maintained and had contamination from nearby septic tanks, sludge systems, and waste dumpsites [58]. An Egyptian study also isolated these bacteria from ground water in rural areas [59]. While efforts to improve personal practice with water, sanitation, and hygiene measures, food

sources and preparation, and animal husbandry are important, exploring the endemicity of enteric pathogens in the environment should also be explored.

This work was conducted in Timor-Leste to address critical gaps in the existing literature and contribute to the understanding of enteric infections and malnutrition in infants and children in the country. Timor-Leste and similar Southeast Asian countries have limited evidence on the prevalence of enteric pathogens in clinical and environmental settings, making it crucial to conduct studies in these regions. Current estimates on malnutrition in LMICs are based on large multicentre studies set in South America, Africa and South Asia. By focusing on Timor-Leste, this pilot work provides valuable insights into the unique conditions and challenges faced in Timor-Leste including risk factors and endemicity of enteric pathogens, which may differ from larger, multicentre studies conducted in different geographic locations. Moreover, conducting current and future research in Timor-Leste allows for the testing of novel hypotheses tailored to the local population, environmental conditions, and sociocultural factors, leading to more targeted interventions and improved health outcomes.

This study highlighted some of the significant research challenges faced in research in small LMIC settings and has several limitations. We had reasonably high attrition of our birth cohort, due to highly mobile families especially during the COVID-19 pandemic when the study was interrupted between April and July 2020 and during flash flooding events. This led to substantial loss to follow up and results that were under-powered to examine some important relationships. To increase study retention, our future study will increase focus on engagement and collaboration with participants and incorporate flexible and frequent follow-up methods. In particular, we will confirm whether the address provided at the hospital is a permanent or temporary residence (e.g., short-term stay with family closer to hospital than permanent residence) and maintain regular contact to update participants should delays occur between visits. Stool samples were not always available for collection at the time of visit and required return visits by the study team. Collecting specimens is an imposition on busy families, can be easily forgotten, requires training in safe handling, and provision of safety equipment. In the future, we will consider using flocked rectal swabs by trained study team members to collect specimens during home visits. These swabs have been relatively successful in other studies compared with traditional stool specimens for pathogen detection rates and bacterial culture [60, 61].

As we recruited all eligible children in hospital with a SAM or severe diarrhoea diagnosis until the target was met, the hospital cohort is likely not representative of children with these conditions in Dili. A future study would attempt to combat this by setting in place more specific demographic targets and include a control population of children diagnosed with other health conditions. We used visual assessment of stool samples by laboratory staff to define diarrhoea in children. We note that visual inspection at the laboratory cannot provide details on the number of bowel movements, total stool volume, or duration of illness, nor can it compare the current sample to normal stool patterns in the infant [62]. Multiplex PCR testing is very sensitive and detects presence of pathogen nucleic acid but not necessarily viable pathogen. The Biofire® GI PCR panel has shown high sensitivity and specificity in detection of diarrhoeal pathogens [63]. We intended to culture all human clinical *Salmonella*, *Campylobacter* and *Shigella* samples at NHL, however due to a lack of available resources this was not completed in most cases. Further, limited animal faecal or environmental samples were able to be collected following a positive PCR or culture result due an outbreak in African Swine fever in the first six months of the study. This limited our ability to estimate the prevalence of enteric infections in the surrounding home environment. In the absence of culture results to compare to the positive PCR results for *Campylobacter*, *Shigella* and *Salmonella* we were unable to determine the validity of the PCR results in this setting. However, similar studies have shown

that multiplex PCR testing can provide an accurate description of enteric pathogen prevalence in low- and middle-income countries [64, 65]. This was pilot study to test acceptance and practicality of research and to build in-country laboratory and research capacity. Due to the funding available we were limited to recruiting from the main referral hospital and its immediate catchment area. Infants born at home or in provincial hospitals were not able to be included which may have led to bias in recruitment of urban or semi-urban families.

We detected high prevalence of enteric pathogens, particularly the bacterial pathogens DEC and *Campylobacter* spp., and a high proportion of wasting from a young age in children in Dili, Timor-Leste. While we faced local challenges, we have identified strategies for mitigating these issues in the future and have produced practical information to guide the development of the next phase of this study. We suggest further exploration of risk factors in a high-powered study with a focus on feeding practices (e.g., exclusive bottle feeding) and the risk of enteric pathogen infection. Further, we recommend greater sampling from animal and environmental sources to identify potential transmission pathways. This would utilise a One Health collaborative approach between hospital and human health laboratory professionals and animal health experts including veterinarians and laboratory specialists to enhance local skills and capacity.

## Supporting information

**S1 Checklist. PLOS inclusivity in global research checklist.**
(PDF)

**S1 Table. Weight-for-height, height-for-age, and weight-for-age z-score for birth cohort infants at each home visit in Dili, Timor-Leste, 2019–2020.**
(PDF)

**S2 Table. Adjusted univariate odds ratios using a generalised estimating equations model for differences in pathogens detected between diarrhoeal and non-diarrhoea stool samples for infants in a birth cohort in Dili, Timor-Leste, 2019–2020.** * Indicates statistical significance (p value <0.05). 95% CI = 95% confidence interval. ref = reference variable. NA = odds ratio not calculated.
(PDF)

**S3 Table. Adjusted univariate odds ratios using a generalised estimating equations model for risk factors associated with pathogen detection for infants in a birth cohort, Dili, Timor-Leste, 2019–2020.** * Indicates statistical significance (*p* value <0.05). GEE aOR = adjusted odds ratio from generalised estimating equation model. 95% CI = 95% confidence interval. ref = reference variable. NA = odds ratio not calculated.
(PDF)

**S4 Table. Adjusted univariate odds ratios using a generalised estimating equations model for risk factors associated with moderate wasting for infants in a birth cohort, Dili, Timor-Leste, 2019–2020.** * Indicates statistical significance (*p* value <0.05). GEE aOR = adjusted odds ratio from generalised estimating equation model. WHZ = weight-for-height z-score. 95% CI = 95% confidence interval. ref = reference variable. NA = odds ratio not calculated.
(PDF)

**S5 Table. Weight-for-height, height-for-age, and weight-for-age, and MUAC-for-age for hospital cohort children at admission, Dili, Timor-Leste, 2019–2020.**
(PDF)

**S6 Table. Mean age in months at admission by diagnosis and type of enteric pathogen detected for hospital-based surveillance cases, Dili, Timor-Leste, 2019–2020.**
(PDF)

**S7 Table. Adjusted univariate odds ratios using a generalised linear model for differences in pathogens detected between diarrhoeal and non-diarrhoea stool samples for children in a hospital surveillance cohort in Dili, Timor-Leste, 2019–2020.** * Indicates statistical significance (p value <0.05). aOR = adjusted odds ratio. 95% CI = 95% confidence interval. ref = reference variable. NA = odds ratio not calculated.
(PDF)

**S1 Fig. Total pathogen detections for a birth cohort and hospital surveillance cohort, stratified by age group, in infants and children in Dili, Timor-Leste, 2019–2020.**
(PDF)

**S1 File. Questionnaire administered to families during birth cohort home visits in Dili, Timor-Leste, 2019–2020.**
(PDF)

**S1 Graphical abstract.**
(TIF)

## Acknowledgments

The authors thank all persons and organisations that provided support for this study. These include: Professor Nélson Martins and the National Institute of Health-Research Ethics & Technical Committee; Dr Dongbao Yu and Dr Rajesh Pandav at the World Health Organization; Dr Merita Monteiro at Departamentu Kontrolu Moras Hada'et (CDC); Maria Angela Varela Niha at Departamentu Vijilansia Epidemiolojia, Ministério da Saúde; Dr Jeremy Beckett at Maluk Timor; local research team members Francisca Soares and Antoninho Gusmão; Menzies staff including Dr Jennifer Yan, Salvador Amaral, Dr Ian Marr, Lucsendar Alves, Dr Jo Wapling, Karen Champlin, and Dr Shawn Ting; Ismael Barreto and all staff from the National Health Laboratory; all staff from the Diagnostics Laboratory for Animal Health; all staff at HNGV including Dr Flavio Brandão de Araujo, Dr Ingrid Bucens, Dr Amelia Ferreria Pinto and the HNGV Paediatrics team; Ms Melanie McVean and the St John of God Midwifery team; the HNGV Obstetrics and Midwifery team, and; Dr Anna Okello, Dr Francette Geraghty-Dusan, and Bethany Lees from the Australian Centre for International Agricultural Research.

## Author Contributions

**Conceptualization:** Kathryn Glass, Joshua R. Francis, Benjamin G. Polkinghorne, Martyn D. Kirk.

**Data curation:** Nevio Sarmento, Almerio Moniz, Benjamin G. Polkinghorne.

**Formal analysis:** Danielle M. Cribb.

**Funding acquisition:** Benjamin G. Polkinghorne, Martyn D. Kirk.

**Investigation:** Nevio Sarmento, Almerio Moniz, Nicholas S. S. Fancourt, Milena M. Lay dos Santos, Endang Soares da Silva, Virginia de Lourdes da Conceição, Feliciano da Conceição, Paulino da Silva, Samantha Colquhoun.

**Methodology:** Danielle M. Cribb, Nevio Sarmento, Nicholas S. S. Fancourt, Kathryn Glass, Anthony D. K. Draper, Joshua R. Francis, Benjamin G. Polkinghorne, Joanita Jong, Martyn D. Kirk, Samantha Colquhoun.

**Project administration:** Danielle M. Cribb, Benjamin G. Polkinghorne, Samantha Colquhoun.

**Resources:** Nicholas S. S. Fancourt, Anthony D. K. Draper, Joshua R. Francis, Milena M. Lay dos Santos, Endang Soares da Silva, Benjamin G. Polkinghorne, Virginia de Lourdes da Conceição, Feliciano da Conceição, Paulino da Silva, Samantha Colquhoun.

**Supervision:** Kathryn Glass, Martyn D. Kirk, Samantha Colquhoun.

**Validation:** Benjamin G. Polkinghorne, Samantha Colquhoun.

**Writing – original draft:** Danielle M. Cribb, Kathryn Glass, Anthony D. K. Draper, Martyn D. Kirk, Samantha Colquhoun.

**Writing – review & editing:** Danielle M. Cribb, Nevio Sarmento, Almerio Moniz, Nicholas S. S. Fancourt, Kathryn Glass, Anthony D. K. Draper, Joshua R. Francis, Milena M. Lay dos Santos, Endang Soares da Silva, Benjamin G. Polkinghorne, Virginia de Lourdes da Conceição, Feliciano da Conceição, Paulino da Silva, Joanita Jong, Martyn D. Kirk, Samantha Colquhoun.

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
