## [Decision Letter · Decision Letter 0]

12 Dec 2023

PONE-D-23-35940A pilot study using hospital surveillance and a birth cohort to investigate enteric pathogens and malnutrition in children, Dili, Timor-Leste.

PLOS ONE

Dear Dr. Cribb,

I am pleased to inform you that one reviewer already endorsed the revised manuscript (although he recommended adding a graphical abstract of the present study) and the other reviewer only requested minor revisions for future publication endorsement. Please carefully answer the first reviewer's concerns and rectify the manuscript following the reviewer's comments. Also, I would recommend you follow the suggestion of the second reviewer.

Thank you and best regards,

António Machado

Note from PLOS Editorial Office: *This manuscript was originally submitted as **PONE-D-23-14515.* *In order for there to be a complete record of the peer review history I have appended the original decision letter and the original comments made by one of the reviewers for PONE-D-23-14515.*

Reviewers' comments:

Reviewer's Responses to Questions

**Comments to the Author**

1. Is the manuscript technically sound, and do the data support the conclusions?

Reviewer #1: Yes

Reviewer #2: Yes

2. Has the statistical analysis been performed appropriately and rigorously? 

Reviewer #1: Yes

Reviewer #2: Yes

3. Have the authors made all data underlying the findings in their manuscript fully available?

Reviewer #1: Yes

Reviewer #2: Yes

4. Is the manuscript presented in an intelligible fashion and written in standard English?

Reviewer #1: Yes

Reviewer #2: Yes

5. Review Comments to the Author

Reviewer #1: I would now recommend publication with minor adjustments (see below). The authors have demonstrated a staggering prevalence of pathogens and malnutrition for the critically understudied Dili Timor-Leste region, and these numbers should be published. The authors have accomplished what one usually sets out to accomplish in pilot studies: though underpowered (as pilot surveys often are), they have provided a foundational evidence base upon which to inform future studies in this area. This is critical for informing local and national governments on areas of focus for research and intervention.

Please see attached file for more specific comments.

Reviewer #2: Excellent work. It is a good idea to raise it as a pilot study. Ah futuro se le puede dar diferentes direccions con el proposito de aprovecharlo al máximo. My suggestions point to the visualization of the study. I would like to have a graphical abstract and add and/or combine some graphs into one.

Regards

FC

6. PLOS authors have the option to publish the peer review history of their article (what does this mean?). If published, this will include your full peer review and any attached files.

Reviewer #1: No

Reviewer #2: **Yes: **Fausto Cabezas Mera

---

## [Author Response · Author response to Decision Letter 0]

14 Dec 2023

We thank the editor and reviewers for the time with our manuscript. We have provided a point-by-point response to reviewers. We have also updated the manuscript and files to conform with PLOS ONE style requirements, and have fixed errors in our financial disclosure statements.

---

## [Editor Report · Decision Letter 1]

19 Dec 2023

A pilot study using hospital surveillance and a birth cohort to investigate enteric pathogens and malnutrition in children, Dili, Timor-Leste.

PONE-D-23-35940R1

Dear authors,

Thank you for answering all suggestions recommended by both reviewers, although both reviewers already had endorsed the manuscript for publication.

Thank you for choosing Plos ONE journal to publish your study.

Best regards,

António Machado
---

## [Editor Report · Acceptance letter]

24 Jan 2024

PONE-D-23-35940R1 

PLOS ONE

Dear Dr. Cribb, 

I'm pleased to inform you that your manuscript has been deemed suitable for publication in PLOS ONE. Congratulations! Your manuscript is now being handed over to our production team.

Kind regards, 

on behalf of

Dr. António Machado 

Academic Editor

PLOS ONE